# Explore Big Data Analytics Applications and Opportunities: A Review

**Zaher Ali Al-Sai** [1,2,*], **Mohd Heikal Husin** [2], **Sharifah Mashita Syed-Mohamad** [2], **Rasha Moh'd Sadeq Abdin** [2], **Nour Damer** [3], **Laith Abualigah** [2,4,5,6,7] **and Amir H. Gandomi** [8,9,*]

1   Department of Management Information Systems, Faculty of Business, Al-Zaytoonah University of Jordan, Amman 11733, Jordan
2   School of Computer Sciences, Universiti Sains Malaysia, Pulau Pinang 11800, Malaysia
3   King Talal School of Business Technology, Princess Sumaya University for Technology, Amman 11941, Jordan
4   Prince Hussein Bin Abdullah College for Information Technology, Al Al-Bayt University, Mafraq 25113, Jordan
5   Faculty of Information Technology, Al-Ahliyya Amman University, Amman 19328, Jordan
6   Faculty of Information Technology, Middle East University, Amman 11831, Jordan
7   Faculty of Information Technology, Applied Science Private University, Amman 11931, Jordan
8   Faculty of Engineering and Information Technology, University of Technology Sydney, Sydney 2007, Australia
9   University Research and Innovation Center (EKIK), Óbuda University, 1034 Budapest, Hungary
*   Correspondence: z.alsai@zuj.edu.jo (Z.A.A.-S.); gandomi@uts.edu.au (A.H.G.)

**Abstract:** Big data applications and analytics are vital in proposing ultimate strategic decisions. The existing literature emphasizes that big data applications and analytics can empower those who apply Big Data Analytics during the COVID-19 pandemic. This paper reviews the existing literature specializing in big data applications pre and peri-COVID-19. A comparison between Pre and Peri of the pandemic for using Big Data applications is presented. The comparison is expanded to four highly recognized industry fields: Healthcare, Education, Transportation, and Banking. A discussion on the effectiveness of the four major types of data analytics across the mentioned industries is highlighted. Hence, this paper provides an illustrative description of the importance of big data applications in the era of COVID-19, as well as aligning the applications to their relevant big data analytics models. This review paper concludes that applying the ultimate big data applications and their associated data analytics models can harness the significant limitations faced by organizations during one of the most fateful pandemics worldwide. Future work will conduct a systematic literature review and a comparative analysis of the existing Big Data Systems and models. Moreover, future work will investigate the critical challenges of Big Data Analytics and applications during the COVID-19 pandemic.

**Keywords:** big data; big data analytics; big data applications; big data opportunities; COVID-19 pandemic; medical applications; healthcare; education

## 1. Introduction

The COVID-19 pandemic has drastically changed nation's worldwide routine life and operations. People have been forced to study and work from home, commuting and traveling to local and overseas destinations have become impossible, and governments have been forced to close cities and countries' borders [1,2].

There is undoubtedly a need for the transfer/exchange the big data systems since decision-makers must be able to react swiftly to changes or trends in markets, investments, interest rates, and other crucial happenings [3]. Decision-makers should be thoroughly aware of the type of inputs they have and the best structure for exchange or analysis if they are thinking about making significant investments in KM systems or big data/business analytics systems [3].

The COVID-19 pandemic paralyzed most vital industries globally. Recently, this is the presented fact; however, a real opportunity is hidden in the new oil in the digital

economy, which is Big Data. From a theoretical point of view, researchers have identified different BD-related capabilities and resources as a solid and potential foundation to enhance organizational performance. Most current works in the BD Analytics (BDA) domain cover the technology dimensions, talent, and management that can impact organizational performance [4]. The organization's ability to benefit from different forms of massive data is highly required, and the willingness to invest in BD is now at the center of interest [5,6].

Recently, it has been normal for organizations to be under pressure to remain in their positions in fiercely competitive markets and identify strategies of expenditure reduction, quality enhancement, and reduced time to market [7]. The new era of BD transformation needs next-generation technologies to attain success [8–13].

Organizations will be required to manage it appropriately for competitive advantage and durability in the modern digital market [14]. Organizations should be capable of identifying vital data resources, structure, needed skills, and architecture. Moreover, organizations must define and describe the underlying infrastructure of the process that supports BD analysis, formulate and applicable BD strategy, and measure applications and technologies that support the organization's requirements regarding their BD investments. Particularly, organizations should migrate their data collection and analysis from just being product or service orientated to a future-oriented platform [14]. To grow the adoption rate, ensure the successful implementation, and minimize the risk after implementation, it is crucial to assess and measure BD readiness and maturity level using a maturity assessment model and tool [15,16].

For instance, in the education sector, big data analytics played a key role in overcoming the negative consequences of the pandemic on the educational sector. It supported tutors and instructors to personalize the remote learning experience for educators. Additionally, it helped bridge the unemployment gap that resulted from COVID-19 major economic losses globally. The importance of big data applications and analytics in the transportation field of COVID-19 has been explicitly shown to decision-makers. For instance, regulators supported their decisions and judgments based on the data captured and analyzed via AI techniques and predictive models. Based on the results, precautionary measures were clearly defined, and any violations were easily detected. Furthermore, predictive models guided decision-makers on citizens' movement within and among cities and metropolitans; consequently, they were able to detect and predict future endemic areas.

Existing literature review papers have covered the topic of BD applications during COVID-19. However, this review paper presents new insights into how big data analytics is integrated into the picture in four critical industries. More specifically, this paper will present how big data applications and their aligned data analytics can pave the road for industries to survive an uncontrolled and unpredictable situation. This paper explains these applications in detail. A systematic comparison between the use of BD applications before and after COVID-19 is presented. The paper focuses on four highly impacted industries: Healthcare, Education, Transportation, and Banking. Additionally, this study analyzes the alignment of big data applications with their relevant data analytics models in the era of COVID-19.

The structure of this review paper will be presented as follows: The introduction Section 1, then the literature review Section 2, which highlights the definition and characteristics of Big Data. Additionally, it highlights the Big Data Analytics and types of analytics. Then big data applications and opportunities are presented in Section 3. Sections 4 and 5 review the Big Data applications before and during COVID-19, specifically. Finally, some future work suggestions will be presented in the conclusion and future work in Section 6.

## 2. Literature Review

Big Data is a critical asset in the competitive market of the digital economy. The benefits of Big Data allow organizations to achieve various objectives under the umbrella of Big Data insights [17]. The following sub-sections present the overall review of Big Data and its applications.

### 2.1. Big Data and Analytics

There has not been a standardized definition for BD among industry, business, media, academia, and various stakeholders. Absence of a systematic definition for BD concept leads to a sort of confusion [12,18,19]. BD is usually defined by individuals. It is different from one industry to another, and according to the types of available sizes of datasets and the software tools are common in a particular industry [8,20–22].

There have been remarkable thoughts from both industry and academia on BD definition [23]. By coupling the concept of BD with current grounded academic research, the BD concept can be more understandable. A clear view of BD concept will enhance the awareness about BD phenomenon for both practitioners and academics, resulting in faster growth and more efficient value obtained from BD [24]. In spite of the fact that there is no identified definition for BD, from a technical and business point of view, BD is identified as the increasing flow of various types of data from different resources [25].

The first BD definition was written by scientists from NASA. The paper published in 1997, by NASA referred to the data volume as an exciting challenge for computer systems to increase the demand for the big volume of main memory, local disk, and in addition to a remote disk. It was identified by NASA as the problem of BD that required to obtain more resources [8–13,26]. The META Group analyst Dough Laney (now Gartner) has defined data growth challenges and opportunities in to three-dimensional (velocity, volume, variety) [18,27].

The researchers have defined BD concepts from different point of views (BD characteristics, technology, business, Innovation, etc.). One of the definitions had been updated by Gartner in 2013, who defined BD concept as "high-volume, high velocity and/or high variety information assets that demand cost-effective innovative forms of information processing for enhanced insight, decision making, and process optimization" [26,28,29]. The Statistical Analysis System Institute (SAS) defined BD as "Popular term used to describe the exponential growth, availability, and use of information, both structured and unstructured" [30]. IBM also added a definition for BD, "Data is coming from everywhere; sensors that gather climate information, social media posts, digital videos and pictures, purchase transaction record, and GPS signal of mobile phone to name a few", "BD can be defined as large set of very unstructured and disorganized data", "BD is a form of data that oversteps the processing power of traditional database infrastructures or engines" [30–32].

BD was referred from more than one perspective (BD as technology, entity, and process) [33]. The definition of BD analytics consists of the technologies (database and data mining tools) and techniques (analytical methods and techniques) that organizations can utilize to analyze vast amount and complex data for a variety of applications prepared to increase the performance of organizations in many perspectives. BD can be considered as both entity and process. BD as an entity includes a volume of data captured from a variety of resources (internal and external) and consists of structured, semi-structured, and unstructured data that cannot be processed using traditional databases and software techniques. BD as a process refers to both the organizations' infrastructure and the technologies used to capture, store and analyze numerous types of data [10–13,33].

New insights are provided by BD to discover new values, supporting organizations to get the benefit of a deep understanding of the hidden values [23]. BD is pointed out as a technology that enables the processing of unstructured data; and BD technologies are the systems and tools used to process BD such as NoSQL databases, the Hadoop Distributed File System, and MapReduce [34,35].

According to [14], different theories and definitions on what shape BD exist in are provided. The most often referred definition is BD oversteps the capabilities of popularly and currently used software tools and hardware platforms to capture, manage, and process it within an acceptable and bounded time. The concept of BD has been promoted to define the novel and powerful computational technologies that have been provided to process an enormous volume of data. BD has been described in various ways, however, fundamentally is a modern technology that is primarily characterized and derived from Business Analytics

(BA) and Business Intelligence (BI). It is capable of creating business values via its predictive analytics, and decision support abilities, which results in the potency to deal with data that traditional techniques cannot process [25,34].

According to the studies by [13,36], BD is defined as "a term that describes large volumes of high velocity, complex and variable data that requires advanced techniques and technologies to enable the capture, storage, distribution, management, and analysis of the information".

### 2.2. Characteristics of Big Data

Existing work characterized BD as novel technologies and architectures which are designed for extracting value from enormous volumes of a wide range of data, by empowering high-velocity capture, discovery, and analysis in a cost-effective way [28]. Since BD is relatively new, it is significant for organizations to know what makes this trend valuable and they should identify the "Vs" that describe the key characteristics of BD [37]. Still, a lot of confusion and obscurity among the Vs of BD exists. Some pioneering studies pointed out that there are three, four, five, and sometimes even seven characteristics of BD [38].

The large-scale feature of BD is reflected in three different characteristics of volume, variety, and velocity. Traditional technologies do not have the ability to successfully deal with the enormous data volume, which is generated at a growing velocity, via online streaming and a variety of other different resources such as transactional systems, sensors, social media, product/service instrumentation, and web platforms [38]. META Group analyst Doug Laney (now Gartner) presented 3Vs of BD to characterize the data management in 3 dimensions represented by three main Vs of Volume, Velocity, and Variety [39]. Volume represents the amount of data. Velocity represents the speed of data generation and process. Variety refers to the diversity of resources and data types. Variety refers to the diversity of resources and data types [40–42]. The three Vs have been mentioned by NIST and Gartner in 2012 and extended by IBM to involve the 4th V representing "Veracity ". Contrarily, Oracle avoided using the paradigm of "Vs" in its BD definition. Instead, it is highly believed that BD is the derivation of values from traditional relational database-driven business decision making, grown with new resources of unstructured data [18].

The 4Vs (volume, variety, velocity and value) model was presented by [14,33,41,43,44]. Excluding the 4Vs mentioned, another V which is veracity is identified to represent the uncertainties of BD and data analysis outcome. Another research conducted by [41,42], pointed out the four major Vs of BD namely volume, velocity, variety, and value that pertains to the insight obtained by organizations from BD which not only require scalability, but also for preferable operational procedures and strategies [41,44–46] pointed out five key characteristics of BD as 5Vs (Volume, Velocity, Variety, Veracity, and Value). "Complexity" is a "C" feature added to the 4-Vs (Volume, Variety, Velocity, Value) of BD by [47–49] to formulate another 5 characteristics of BD. Security and management are additional characteristics to the 3Vs (Volume, Variety, and Value) [48]. A study by [48] also presented a critical problem of technical research that requires more investigation by scholars.

Recently, other Vs which are (Visualization/Visibility, Variability/Volatility, Validity, Virtual, and Complexity) are added to BD characteristics by [26]. Another work done by [50], defined the 7 Vs of BD namely Volume, Velocity, Variety, Value, Veracity, Variability which implies inconstancy and heterogeneity; and visualization which implies the illustrative character of data. Volume, Velocity, Variety, Veracity, and Value are the widely accepted and common Vs by stakeholders. However, the other Vs are important for BD paradigm too. By comparing existing definitions of BD and its related aspects, the 5Vs (volume, velocity, variety, veracity, and value) characteristics are extracted and formulated to point out how different traditional data and BD are [24,51,52] as illustrated in Figure 1.

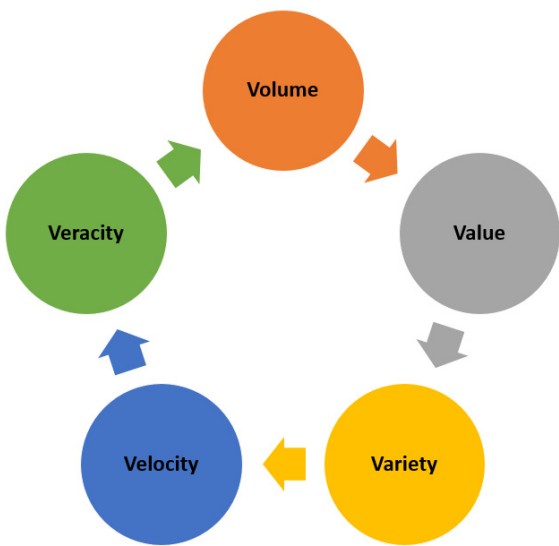

**Figure 1.** The Five Features of Big Dat.

Big data is more of a concept than an exact term. Some classify big data as a volume problem only for petabyte-scale (>1 million GB) data collection. Some people associate big data with different data types, even if the volume is measured in terabytes. These interpretations made the big data problem situational [51].

*2.3. The Types of Data Analytics*

Big Data Analytics refers to the process of collecting, organizing, and analyzing high volume, velocity, variety of data to discover the valued patterns that could use for making decisions. Analyzing the big data need new tools, methods, and technologies such as data mining, predictive analytics, and perspective analytics [52].

Most of existing literature identified the use of big data applications defined in the presence of the four types of data analytics. The four types of big data analytics that can be implemented in governments are: (i) Descriptive, (ii) Diagnostic, (iii) Predictive, (iv) Prescriptive [53].

The following section will describe each type with their related examples in governments and more specifically during COVID-19:

- Descriptive

Descriptive analytics is the preliminary stage in the analytics categorization. Descriptive analytics is known as business reporting, as such stage emphasize in creating summary reports to highlight business activities, and to illustrate the answers of questions of "what is happening or happened?" [54–56].

This type of data analysis depends on analyzing past data, visualize and understand historical trends. An example of BDA during COVID-19 are Dashboards used in the health care sector to monitor live data about the spread of COVID-19 in a particular area. Such dashboards track, illustrate and statistically explain the historical records captured about COVID-19 cases in a specific area, city or country [29].

- Diagnostic

The second data analytics type is Diagnostic data analysis. This type focuses on illustrating the correlation, hidden patters, cause-effect relation and interrelationships between different variables. An example would be the data captured from job portals. Such data is used to analyze and visualize potential market sectors and match it with the relevant workforce in the country [54].

Diagnostic analytics figure out answers to questions of "why did it happen?'. The main goal in Diagnostic Analytics is to highlight the root causes of a challenge or problem.

Such root causes identification depends on specialized techniques such as visualization, drill-down, data discovery, and data mining [54–56].

- Predictive

Predictive analytics is categorized in the third level on the data analytics hierarchy. More specifically it is the stage residing after the descriptive analytics. Based on the Data Analytics maturity model, organizations that have matured in descriptive analytics can move forward to the next stage to answer "What will happen?" [55,56].

The third data analysis focuses on patterns from past existing data and predict what will happen when changes occur in such set of data. The example here is the vaccine distribution prioritization mechanism. Data analysts predict through machine learning model who is next in need to the vaccine and prepare patient priority list accordingly [57].

- Prescriptive

The last data analysis type Prescriptive analytics. It is where the best alternative among many–that are usually created/identified by predictive and/or descriptive analytics– courses of action is determined using sophisticated mathematical models. Therefore, in a sense, this type of analytics tries to answer the question of "What should I do?". Prescriptive analytics uses optimization, simulation, and heuristics-based decision modelling techniques [58].

Perspective analytics is ranked as the highest level in data analytics maturity model, it is also viewed as the most sophisticated and complex data analysis type. Both AI and big data analysis techniques are used in Prescriptive analytics. Utilizing such techniques facilitate decision makers to frame the optimal strategic decision. Decision makers will reach to these decisions via selected optimization models. For example, Prescriptive analytics were used during COVID-19 pandemic to understand citizens' reactions towards the vaccine, and support decision makers to structure the optimal strategic decisions to control citizens' hesitant towards the vaccine [55,56].

To illustrate how the four data analytics types are classified based on the level of sophistication and data complexity, researchers have introduced the following two categorize as shown in Figure 2:

- Business Intelligent, which consist of both Descriptive analytics and Diagnostic analytics
- Advanced Analytics, which consider the higher data analytics types in maturity level, namely predictive and prescriptive analytics [55,56,59].

| Business analytics | | |
|---|---|---|
| **Descriptive** | **Predictive** | **Prescriptive** |
| **Questions** | | |
| What happened?<br>What is happening? | What will happen?<br>Why will it happen? | What should I do?<br>Why should I do it? |
| **Enablers** | | |
| Business reporting<br>Dashboards<br>Scorecards<br>Data warehousing | Data mining<br>Text mining<br>Web mining<br>Machine learning | Optimization<br>Simulation<br>Decision modeling<br>Network science |
| **Outcomes** | | |
| Well defined business<br>problems and opportunities | Accurate projections | Best possible business<br>decisions and actions |

**Figure 2.** A simple taxonomy for analytics.

### 3. Big Data Analytics Opportunities and Applications

Big data analytics can be described as the use of mathematical and statistical techniques, to find the hidden patterns and variances in large amount of data from multiple sources, and from different type of data (structure, semi-structured, unstructured) to gain future insight and faster decision making [60]. Such findings will be the base for organizations to provide them with valuable knowledge and support them in their strategic decisions [61]. The utilization of big data analytics has shown an added value to governments and firms during COVID-19. Consequently, those who have implemented big data analytics, outperform others. For instance, they were able to map their current status and structure better strategic decisions [55].

In a Mckinsey's report it was highlighted that big data analytics empowered those who applied it, by incrementing their annual economic value between $9.5 trillion and $15.4 trillion [62]. Furthermore, as the COVID-19 outbreak, big data analytics has emphasized its effectiveness in detecting the spread of COVID-19, and supported governments to reach optimal decisions against it [63].

The main goal for organizations is the bottom line represented in their profits, market share and customer loyalty and satisfaction levels [64]. This fact is applied for both business firms and governmental entities. With the exponential increase in the volume of data, the speed in which it is generated, the variety of sources generating it, and the importance of its quality and relevance. The vital role of big data applications in various business sectors and governmental entities have been a necessity for their success [65]. The implementation of big data applications has supported organizations to enhance their customers experience, improve cost savings, and facilitate strategic decision making [66]. Consequently, organizations' processes and operations become achieve a higher level of effectiveness and efficiency [67].

New, advanced and tactical digital technologies were considered recently as a response to the COVID-19 pandemic, such as big data applications. Countries such as Taiwan, South Korea, Hong Kong, and Singapore have demonstrated the significant positive impact from adopting such applications. Those countries proved the seamless of controlling the pandemic expected risks effectively [61].

Big Data Applications can derive insights from various data sources to provide ideal solutions for several sectors [52]. Organizations from a variety of industries have started using MapReduce-based solutions for processing enormous amounts of data [68]. To meet their needs for handling large-scale data processing, many businesses rely on MapReduce. As businesses from a variety of sectors embrace MapReduce together with parallel databases. new MapReduce workloads have appeared that contain a large number of brief interactive tasks [68].

Table 1 highlights the alignment of each big data application to its respective big data analytics model. It provides an explanation on how such an implementation has supported organizations and governments to cope with COVID-19 pitfalls. Furthermore, such an implementation provided an optimal solution to harness its operations and decision-making process. This categorization has been developed based on the description of each application in their relevant fields, and on the definition highlighted in the section on the four big data analytics types.

**Table 1.** How to utilize Big Data Analytics in Healthcare, Education, Transportation, and Banking.

| Field | Data Analytics Type | How BDA Has Been Utilized | Data Processing Models Used to Analyse Big Data | Reference |
|---|---|---|---|---|
| Healthcare | Descriptive and Predictive Data Analytics | Proactive actions and interventions based on predictive models to trigger any noncommunicable diseases. | Predictive models based on search engines and social media data. Smart phone applications tracking system to identify infection hot spots | [69,70] |
| | Perspective Data Analytics | Vaccine distribution | Sentiment analysis to reduce community resistance towards the vaccine. | [69–71] |
| | Diagnostic and Predictive Data Analytics | Vaccine distribution | Machine learning models to prioritize the citizens' need and urgency to the vaccine | |
| | Diagnostic and Prescriptive Data Analytics | Monitoring live and frequent data on the spread of the disease Provide more personalized consultations by "virtual doctors" | Dashboards AI Chabot | [72,73] |
| Education | Descriptive Data Analytics | Enhance online educational platform experience | Analyzing data captured from online educational platforms can ease educators remote leaning experience | [69,74] |
| | Diagnostic Data Analytics | Bridge the gap of unemployment | Analysis of data captured from job portals | [69,75] |
| Transportation | Descriptive and Prescriptive Data Analytics | implementation of precautionary measures-Ensure social distancing in public transportation | Capturing relevant data and use machine learning techniques to detect incompliance actions | [76] |
| | | Detect citizens' commute route to store their travel history. | Use both AI and Big data applications to capture, track and predict valuable insights about citizens movement within and across cities and countries | [77] |
| Banking | | Fraud Detection | Use AI and ML techniques to describe and detect real-time abnormal activities and online transaction, and build ML models based on classification algorithims to predict any suspecious case. | [78] |
| | Descriptive and Predictive Data Analytics | Risk Assessment | Use both diagnositic and prescriptuve data analytics models to analyze real-time data and asses the creditworthiness to customers. Consequenlty developing the appropriate cutomer portfolio and tailor clients needs to their services. Cossequently boosting customers' satisfaction, loayality and enhance banks botom line records. | [78] |

## 4. Big Data Applications Pre the COVID-19 Pandemic

In the following section, a demonstration of how big data applications have been applied, and the opportunities captured from it will be illustrated. The section focuses on certain fields before COVID-19, such as healthcare, education, transportation, and banks.

### 4.1. Big Data in Healthcare

The secret behind utilizing Big Data in the healthcare segment is its powerful ability to highlight the correlation and patterns between different variables rather than finding the casual inference between them. Hence, its capability to predict for the future, and therefore facilitating the e government health sector in its decision-making process [79]. For example, it can support building predictive models for risk and resource use, study the behavioral patterns for patients, analyze the population health, facilitate diagnostic and treatment decisions, use medical images as an input to the clinical decision support system [80]. Assure the safety of the drug and medical devices use on patients and serve individuals health better through analyzing, predicting and monitoring the disease patterns [81].

The sources of data in the healthcare field have elevated exponentially, ranging from the records captured from public hospitals, drug research studies, pharmacists, pathologists, medical laboratories and radiologist. Furthermore, recently other indirect sources are considered such as vital sources such as medical newsletters, websites, social media platforms, health reports, and discussion forums. Additionally, mobile phone applications such as medical smart watches can be considered in the big data process in the healthcare segment [80]. All of these sources of data are the fuel used to enhance performance in health institutes such as in vaccinations, cure of diseases, insurance procedures, and hospital management operations [82].

Big data utilization in healthcare is focused on delivering superior value to individual patients rather than on delivering analysis on general disease cases and volumes of data [29, 80]. Hence, the main goal of big data applications in the health care industry is to serve efficiently considering both value and costs to individual cases [83].

As explained in this section, the use of BDA has been emphasized throughout the medical procedure cycle, affecting the various stakeholders. From patients, medical providers, medical insurance entities, and medical researchers [29].

### 4.2. Big Data in Education

The educational system is one of the main civilization pillars. Its development can characterize the advancement level of any society. Big data has played a vital role in restructuring the educational system. It enabled educational institutes and professionals to personalize the educational experience for students [84]. The main goal for educators and trainers is to provide a high-quality educational scheme and teaching system. This can only happen by understanding that each student has different way of learning, level of competence, readiness to learn, and interests [85]. How this process can be personalized? Big data is the answer. Big data can analyze, find correlation between the data, highlight patterns, provide insights and predict for the ultimate teaching-learning process. Hence, educators and professionals in the educational field, will provide intelligent decisions to enhance the educational regime [86].

Sources of data in the educational field can be categorized into three main levels Micro-level data, Meso-level data, and Macrolevel data [87]. Micro-level data or what is known as clickstream data, consists of the interactions between millions of learners and their learning environment [88]. This includes the learners' interaction with the virtual gamification, simulations, online platforms, and intelligent tutoring systems. Such actions can predict students' interests [87]. Meso-level data (text data) predict the cognitive ability of students through analyzing the computerized text-oriented writing activities [87]. Such data will be analyzed through NLP techniques [89].

Macrolevel data (institutional data) are sets of data that are captured once a year and represents students' demographics, all educational institutes relevant data (admission data, courses enrolled in, major prerequisites) [90]. Done

When considering big data applications in the educational field, all the above categorization of data sources will overlap. For instance, students' interactions through social media represents both microlevel (duration spent, location of the student) and their meso-level (written posts and text-oriented interaction) [87]. Another example is specific simulation games offered by the educational institute, were the three levels of data will be represented miso/micro/and macro [91].

Applying big data techniques have unleashed several opportunities. For example, big data can improve the learning process, by optimizing the selection, of the prior teaching techniques and newly proposed ones to meet the student actual needs and interests [92]. In addition, big data can facilitate choosing the best bundle of resources, tools, and skills that of higher priority to each teaching-learning case, away from human subjectivity [93]. Moreover, big data can support educators in providing real time feedback and construct development plans based on the student interaction within the virtual learning environment [87]. Furthermore, big data can facilitate constructing a more personalized learning environment. It can track, analyze and predict every action and interaction taken by the students in their virtual environment. By collecting data on students' preferences, performance and results, a more comprehensive picture of the student will be developed. For instance, every click in the virtual learning environment can ease the prediction process [94]. This can be tracked from their interests, their doubts, the time spent in each program, their grades, and their preferred learning style [87]. Thus, big data will result in a more satisfactory learning environment for the students [95].

Enriching the learning environment through understanding the student actual performance level, areas of improvement and difficulties. For instance, big data can detect the questions which the student may fail in or struggle on solving it. Hence, big data can generate progress metrics to provide in depth analysis of the student performance [87]. Furthermore, big data provide a great tool to predict the students who may pass successfully or fail. Another crucial opportunity of applying big data in the educational field is to utilize it in the marketing research, were educational institutes can attract outstanding students [86].

*4.3. Big Data in Transportation*

The phenomena of big data application have raised significantly in the transportation field, as a result of the endless flow of both mobility and city data that resides in digital repositories, remote and in situ sensors and mobile phones and captured accordingly in vast volumes and velocities [96]. These data are the base for researchers, economists and regulators to analyze traffic flow, congestion and their social, economic and environmental impacts [97]. Moreover, applying a combination of new methods of analysis such as artificial intelligent approaches, paves the way for predicting and providing innovative solutions for the future. Hence, creating a new revolution in big data in the transportation field. [30].

For example, big data plays a vital role in predicting the cause effect relation between the driving restriction policies and traffic congestion [98]. Big data through its predictive capabilities and the incorporation of economic insights can exceed the ability to understand and analyze the past and real time data, to predict the optimal legislations for traffic congestion issues in smart cities [99].

To understand how big data applications used in transportation, we will illustrate the categorization of various sources of data:

1.    First source of data which is the primary source is the direct physical sensing. Represented, in road-side static sensors such as LiDAR, microwave Radars, and sensors that measure speed, noise, and traffic flow known as acoustic sensors [100]. Other examples are the use of mobile phone technologies such as GPS, GSM, and Bluetooth [97].

2.  The second source of data is the social media sources "human & social Sensing" highlighted in the use of motorists to the smartphone-compatible platforms [101]. For instance Instagram, twitter and others [97].

3.  The third category of data source is urban sensing which is generated by transportation operators. In this category data captured can analyze urban mobility in terms of congestion and traffic flows [102]. This can be performed via credit cards and smart cards scanned through urban sensors from public transit, retail scanners and digital toll systems [97].

*4.4. Big Data in Banking*

In recent years, the massive use of information technology and more specifically, big data, has reshaped the banking sector intensely. This has been remarked by the introduction of digital banking operations and virtual banking systems [103]. The banking sector is a highly competitive environment. To survive in such a competitive environment a proactive strategy and better strategic decisions must be adapted by management. Big data applications are utilized in the banking sector and supported by data mining techniques to transfer customer semi-structured and un-structured data into meaningful insights and derive the ultimate strategic decisions. Such decisions can support banks to increase customer satisfaction, detect fraud cases, ease the merge and acquisition operations [95], optimize banking supply chain performance [104], outperform annual profits and expand market share. An example of how big data applications harness strategic decisions and meet strategic goals is through applying sophisticated algorithms to categorize clients and group them into clusters based on the analysis and interpretation of clients' behaviors. Such technique can facilitate banks to provide valued and satisfactory services to different clients' categories. Moreover, the ability of big data applications to integrate internal and external sources play a vital role in detecting fraud activities [105]. Furthermore, the capability of big data to analyze, predict, and visualize both external market conditions and internal clients' trends and preferences can empower management in considering the ultimate decision to invest in new markets, hence increasing their market share and enhancing profitability [106].

Table 2 depicts a summary of the Big Data opportunities. Moreover, Table 3 provides examples of Big Data applications in certain fields before COVID-19.

**Table 2.** The Big Data Opportunities before COVID-19 Pandemic.

| Field | Opportunities | Description | Reference |
|---|---|---|---|
| Healthcare | Serve efficiently considering both value and costs to individual cases | BDA have powerful ability to highlight the correlation and patterns between different variables rather than finding the casual inference between them and serve individual patients' cases. | [79–81,83,107] |
| Education | Improve the learning process<br>Provide real time feedback and construct development plans<br>Construct a more personalized learning environment<br>Enrich the learning environment<br>Utilize BDA in marketing research purposes for institutions | BDA enables educational institutes and professionals to personalize the educational experience for students | [84,86,87,92–95,108] |
| Transportation | The base for researchers, economists and regulators to analyze traffic flow, congestion and their social, economic and environmental impacts.<br>Apply a combination of new methods of analysis such as AI approaches, to pave the way for predicting and providing innovative solutions for the future in the field of transportation. | BDA predictive capabilities and the incorporation of economic insights can exceed the ability to understand and analyze the past and real time data, to predict the optimal legislations for traffic congestion issues in smart cities. | [30,96–99,109] |
| Banks | Detect fraud cases<br>Ease the merge and acquisition operations Optimize banking supply chain performance<br>Interpret clients' behaviors.<br>Provide valued and satisfactory services to clients.<br>Analyze, predict, and visualize both external market conditions and internal clients' trends and preferences<br>Increase market share and enhance profitability. | BDA supported the introduction of digital banking operations and virtual banking systems | [103,105,106] |

**Table 3.** Examples of Big Data applications by field before COVID-19.

| Field in Charge | Application Name | Description | Reference |
|---|---|---|---|
| Health | Ebola Open Data Initiative | West Africa-data has been utilized to develop an open-source global model for tracking the cases of Ebola cases in in 2014 | [29,110,111] |
| | HealthMap | a platform used to visualize diseases trends and provides an early trigger on the proper response | [110,112] |
| | Proactive listening, mobile phone-based system | Brazil-to govern the issue of bribes in the health services, and handle any related issues and take an immediate and effective action against corruption. | [110] |
| Education | ENOVA | Mexico, through the utilization of data and data analytics can analyze and predict students' interactions. Consequently, boosts the educational strategies and enhances the used tools and techniques in the teaching-learning process. | [113,114] |
| | (PASS) Personalized Adaptive Study Success | The Open University Australia-Predicts course material, beside a more personalized studying environment. The predictive data analytics model is based on analyzing students, individual characteristics, beside other student related data captured from other systems.<br>The main goal of the application is to develop a more customized environment that ensures students involvement, engagement, and retention in an e-learning environment. | [115] |
| Transportation | OpenTraffic platform | An application to support in urban infrastructure decisions, based on data captured from both vehicles and smartphones, to analyze it and visualize it into both historic and real-time traffic situations. | [110,116] |
| | | Seoul, South Africa-the application is used to support night bus drivers to ease their journey from origin to destination. This will occur through capturing data from tremendous number of calls and text data points, as well as private and corporate taxi data sources. | [110] |
| Banks | Avaloq, Finnova, SAP, Sungard and Temenos | OCBC is the largest bank in terms of market capitalization in Singapore. It operates in more than 15 countries globally. It is a success example of the utilization of BDA. For instance, the bank responded to customer actions, customers' personalized events and their demographic profiles. Hence, OCBC Bank succeed in achieving higher customer engagement and increasing the level of customer satisfaction by 20% in comparison to a control group.<br>These core banking applications, such as Avaloq, Finnova, SAP, Sungard or Temenos for example, were designed to handle large amounts of transactions in back-office processes for basic financial products and services, such as bank accounts, deposits, etc. | [104,117] |

## 5. Big Data Applications Peri the COVID-19 Pandemic

In 2020, the world has been experiencing a critical pandemic, COVID-19. Most governments were not expecting such a drastic change in their citizens daily routine and life. From cities in lock down, individuals' quarantine, people working from home to the emphasize on online services [97,118]. Moreover, many of the government's portals, e services and applications were not up to the required standard to fight against such a disaster. That triggered the importance of data analytics and the utilization of Big Data Applications. Many governments were forced to react in a short period of time [97,119]. A variety of applications have been introduced in different fields, to ease individuals' lives, support governments decisions and control the pandemic effect globally [120]. Furthermore, big data analytics, have facilitated governments to embrace remarkable strategic decisions efficiently and effectively. Moreover, data analytics proved its importance in predicting and managing risks associated to supply chain safety, and to external economic, social and legal risks [55,118].

According to a study by EY in 2021, governments are reconsidering the importance of big data to overcome the pitfalls of the pandemic and to recover from it. For instance, nations all around the globe have invested in a visionary action towards utilizing BDA. Example of countries such as, Hong Kong, US, Switzerland, and India [121].

In 2020, a study by United Nations illustrated some of the most important applications used worldwide during COVID-19. The main goal for governments was to share reliable and transparent information about COVID-19, to enhance citizens' awareness about the situation and allow policy makers to plan appropriate actions accordingly. This has been empowered by dashboards, such as in Vancouver and Australia, to track the number of cases and allocate the required community resources accordingly [122]. Furthermore, to ensure social distancing, governments in India and New York has urged their citizen to rely more on online services such as online parking payment in New York City and e- Doctor tele-video consultation to prevent crowds in hospitals in India. Also, China monitored its citizens commuting to work, grocery stores, and shopping malls via the QR health code. Many other BDA were offered by governments such as platforms for e leaning wither for schools or universities. Not to mention the countless services offered for entertainment online for citizens in quarantine periods [72].

The following sections describe the utilization of big data applications during COVID-19 pandemic in four vital sectors: Healthcare, Education, Transportation and Banking. It highlights the significant results approached in each sector and its role during COVID-19 pandemic.

### 5.1. Big Data in Healthcare

The first application of BDA in health care sector post COVID-19, is to support government agencies to detect a specific disease in particular area, monitor the health condition of the citizens and provide a preventive action accordingly. All of such actions will be based on predictive models which will be supported by input from smart phone connected to thermometers and tracking systems, search engines and social media data [118].

Another use of BDA post COVID-19 is the management and control of vaccine distribution. For example, governments will easily understand the community reaction towards the vaccine [71]. This will be figured through applying sentiment analysis to the data captured from social media platforms, and develop strategies that will control the community resistance towards the vaccine [69]. Furthermore, applying machine learning techniques will anticipate and prioritize who will be more vulnerable to the disease, and who should be provided with the vaccine first. Also, BDA guarantee a better technique to store the vaccine. This will be through monitoring the optimal temperature level [72].

Finally, big data solutions helped governmental entities in their vaccine distribution efforts. For instance, big data facilitated in the storage mechanism of the vaccine, since they were kept and stored within precise temperature range. Hence, ensuring the quality level of the vaccine won't be affected by any environmental circumstances through the

distribution chain. Furthermore, machine learning was applied to highlight analyses of the populations. For example, a categorization of the population with health vulnerabilities were easily distinguished. Consequently, a prioritization mechanism and plan for vaccine delivery was prepared [69]. Also, sentiment analysis on citizens' casual conversation on social media were performed by governmental entities. Such large text-data, helped in understanding the public view on immunization. As a result, governments were able to develop the proper communication strategies, to persuade citizens about vaccinations and overcome any hesitancy from it [123].

### 5.2. Big Data in Education

COVID-19 has forced many schools, and educational institutes to shift their physical presence to online educational platforms [74]. BDA supported educators via analyzing data captured from such platforms, to analyze and predict students current and future learning abilities and develop the educators' teaching styles accordingly [69]. Furthermore, an instant need to bridge the gap of unemployment required the implementation of BDA. This has been evident through analyzing the data captured from job portals, communicating the job market needs to educators and thus develop the appropriate curriculum, and communicating it to the targeted segments [75].

### 5.3. Big Data in Transportation

COVID-19 has stimulated governments to reconsider its transportation decision management systems and empower it with big data applications [77]. For instance, Dubai is a leading example in the use of big data applications to ensure social distancing between bus passengers and detect any incompliance. It invested both in using AI and big data applications to capture relevant data such as, data and time of the trip, driver details, the frequency of the vehicle incompliance, and the route number. Hence facilitate applying disciplinary actions accordingly [76].

Another example is to detect citizens commute route in order to store their travel history. This will ease the regulators to detect whom infected patients with COVID-19 virus have contacted. Hence, government regulators can predict which areas might be potentially more affected than others. Therefore, preventive actions will be taken on a more methodical base. Therefore, in a broader context, countries can predict the flow of infected citizens between cities and countries and consequently declare travelling constraints and guidance [78].

### 5.4. Big Data in Banking

The incident of the global COVID-19 pandemic has exponential increased banks' clients use for online transactions using big data applications [124]. A study prepared by the world bank on 29 June 2022, depicted global financial figures as follow: around 76% of adults created personalized accounts wither with financial institutes or mobile money providers compared to 51% in 2011. Also, the increase has been applied to the use of digital payments. For instance, since the hit of the pandemic more than 80 million adults conducted their first digital purchase and payment in India, and more than 100 million adults in China [125]. Additionally, Big data applications and Analytics crucial role was evident to bankers in their strategic and daily operations during COVID-19. Examples in banking are the use of descriptive and predictive data analytics models in Fraud Detection, and the diagnostic and prescriptive data analysis models such as in Risk Assessment [78]. Moreover, financial intermediaries use AI-based systems for fraud detection and analyze the degree of interconnectedness between borrowers, which in turn allows them to better manage their lending portfolio [126]. Banks are increasingly using big data and analytics to assess the creditworthiness of prospective borrowers and make underwriting decisions, where both functions at the core of finance [126].

*5.5. Big Data Analytics across Industry*

Big Data Applications can derive insights from various data sources to provide ideal solutions for several sectors [52]. Table 4 highlights the alignment of each big data application to its respective big data analytics model. It provides an explanation on how such an implementation has supported organizations and governments to cope with COVID-19 pitfalls. Furthermore, such an implementation provided an optimal AI solution to harness its operations and decision-making process. This categorization has been developed based on the description of each application in their relevant fields, and on the definition highlighted in the section on the four big data analytics types.

**Table 4.** How to utilize Big Data Analytics in Healthcare, Education, Transportation, and Banking.

| Field | Data Analytics Type | How BDA Has Been Utilized | Method/Model | Reference |
|---|---|---|---|---|
| Healthcare | Descriptive and Predictive Data Analytics Models | Proactive actions and interventions based on predictive models to trigger any noncommunicable diseases. | Predictive models based on search engines and social media data. Smart phone applications tracking system to identify infection hot spots | [69,70] |
| | Perspective Data Analytics | Vaccine distribution | Sentiment analysis to reduce community resistance towards the vaccine. | [69,71] |
| | Diagnostic and Predictive Data Analytics Models | Vaccine distribution | Machine learning models to prioritize the citizens' need and urgency to the vaccine | |
| | Diagnostic and Prescriptive Data Analytics Models | Monitoring live and frequent data on the spread of the disease Provide more personalized consultations by "virtual doctors" | Dashboards AI Chabot | [72,73] |
| Education | Descriptive Data Analytics Model | Enhance online educational platform experience | Analyzing data captured from online educational platforms can ease educators remote leaning experience | [69,74] |
| | Diagnostic Data Analytics Model | Bridge the gap of unemployment | Analysis of data captured from job portals | [69,75] |
| Transportation | Descriptive and Prescriptive Data Analytics Models | implementation of precautionary measures-Ensure social distancing in public transportation | Capturing relevant data and use machine learning techniques to detect incompliance actions | [76] |
| | Descriptive and Predictive Data Analytics Models | Detect citizens' commute route to store their travel history. | Use both AI and Big data applications to capture, track and predict valuable insights about citizens movement within and across cities and countries | [77] |
| Banking | | Fraud Detection | Use AI and ML techniques to describe and detect real-time abnormal activities and online transaction, and build ML models based on classification algorithims to predict any suspecious case. | [78] |
| | | Risk Assessment | Use both diagnositic and prescriptuve data analytics models to analyze real-time data and asses the creditworthiness to customers. Consequenlty developing the appropriate cutomer portfolio and tailor clients needs to their services. Cossequently boosting customers' satisfaction, loayality and enhance banks botom line records. | [78] |

## 6. Conclusions

The unstable status resulting from COVID-19 forced organizations to realize the real importance of big data applications. It has been evident during pandemics that Big Data adoption enables decision-makers to make smarter decisions in real time. The technologies behind Big Data support organizations to gain valuable insights from their data. Big Data facilitates transforming organizations' practices to a new generation of digital services ensuring that added value for customers will be achieved. Organizations utilize Big Data to detect and analyze the trends and patterns of people's behavior on social networking. Hence, an organization's decision-makers can provide optimal decisions and better, effective, and efficient services and products for the public. This review paper investigated the existing literature to define Big Data, and the types of Analytics, and compared the Big Data applications before and after COVID-19. The comparison was supported by examples from four vital sectors in the industry of Healthcare, Education, Transportation, and Banking as examples of sectors affected by COVID-19. The paper presented a detailed description of the role of data analytics and its alignment with specific big data applications in those fields. Such applications supported organizations and nations to navigate through the COVID-19 pandemic confidently. Hence, they could not only overcome challenges but also unleash opportunities and create value. The limitation of this paper is related to the limited previous studies that investigated the applications and opportunities of big data during the COVID-19 Pandemic. The future work will start by investigating the challenges faced by organizations on different levels, it will also investigate the critical success factors of Big Data and their categories toward developing a conceptual model for Big Data implementation.

**Author Contributions:** Conceptualization, Z.A.A.-S., M.H.H., S.M.S.-M., R.M.S.A., N.D., L.A. and A.H.G.; methodology, Z.A.A.-S.; formal analysis, Z.A.A.-S.; writing—original draft preparation, Z.A.A.-S., M.H.H., S.M.S.-M., R.M.S.A., N.D., L.A. and A.H.G.; writing—review and editing, Z.A.A.-S., M.H.H., S.M.S.-M., R.M.S.A., N.D., L.A. and A.H.G.; visualization, L.A.; supervision, L.A. and A.H.G.; project administration, Z.A.A.-S., M.H.H., S.M.S.-M., R.M.S.A., N.D., L.A. and A.H.G. All authors have read and agreed to the published version of the manuscript.

**Funding:** This research received no external funding.

**Institutional Review Board Statement:** Not applicable.

**Data Availability Statement:** Not applicable.

**Conflicts of Interest:** The authors declare no conflict of interest.

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
