# Peer review of "Explore Big Data Analytics Applications and Opportunities: A Review"

_2504-2289, doi:10.3390/bdcc6040157_

Round 1
Reviewer 1 Report
This paper reviews the existing literature specializing in big data applications pre and peri-COVID-19. Specifically, four industry fields including healthcare, education, transportation and banking are compared. Moreover, this paper presents a detailed description of the role of data analytics and its alignment to specific big data applications. The paper concludes that ultimate big data applications and relevant data analytic models can overcome challenges brought by epidemics and create value. However, it should be further improved in the following aspects.
1. The abstract is too long, which can be more concise, highlighting the significance of this work.
2. The authors are suggested to explain why they chose these four industrial fields including healthcare, education, transportation and banking for analysis.
3. In the last paragraph of the introduction section, section (4) is not introduced. Should "section (3)" be changed to "section (3) and (4) to review the Big Data applications before and during COVID-19, specifically"? Moreover, the authors are also suggested to highlight what you have done in this work.
4. The authors are suggested to compare the relevant data before and during COVID-19 to illustrate the impact of big data analysis in the corresponding fields.
5. The article “Machine Learning for 6G Wireless Networks: Carry-Forward-Enhanced Bandwidth, Massive Access, and Ultrareliable/Low Latency” (IEEE Veh. Technol. Mag. 2020) introduces a lot of state-of-the-art techniques based on AI/ML to support ultra-broadband, ultra-massive access, ultra-reliable and low-latency services in the future networks. The authors are suggested to survey more approaches that how to introduce the AI/ML-based techniques into big data analytics.
Author Response
Dear Reviewer,
Thank you for reviewing our article. Your suggestions have tremendously upgraded the readability and quality of my paper. I am grateful for your recommendations and valuable comments.
- The abstract is too long, which can be more concise, highlighting the significance of this work.
Response: The abstract had been updated and shorten
- The authors are suggested to explain why they chose these four industrial fields including healthcare, education, transportation, and banking for analysis.
Response: Thank you for your valuable comment. We selected these four industrial fields as sectors that had been affected by Big Data and Analytics.
- In the last paragraph of the introduction section, section (4) is not introduced. Should "section (3)" be changed to "sections (3) and (4) to review the Big Data applications before and during COVID-19, specifically"? Moreover, the authors are also suggested to highlight what you have done in this work.
Response:
The last paragraph had been updated
The contributions of this paper had been highlighted
- The authors are suggested to compare the relevant data before and during COVID-19 to illustrate the impact of big data analysis in the corresponding fields.
Response: Sections 4 and 5 demonstrate how big data applications have been applied, and the opportunities captured from them had been illustrated. The section focuses on certain fields before COVID-19, such as healthcare, education, transportation, and banks.
- The article “Machine Learning for 6G Wireless Networks: Carry-Forward-Enhanced Bandwidth, Massive Access, and Ultrareliable/Low Latency” (IEEE Veh. Technol. Mag. 2020) introduces a lot of state-of-the-art techniques based on AI/ML to support ultra-broadband, ultra-massive access, ultra-reliable and low-latency services in the future networks. The authors are suggested to survey more approaches that how to introduce the AI/ML-based techniques into big data analytics.
Response: Thank you for your valuable comment and recommendation. Future work of on the progress paper will investigate the approaches of AI/ML-based techniques into big data analytics.
Kindly find the response to the comments you suggested in our paper.
Thank you so much,
Reviewer 2 Report
The subject of this article is very interesting.
After a very attentive analysis, I have the next recommendations for the authors of this article:
1. The abstract must contain very clearly the objective, the methodology applied, the results on every four fields, and originality;
2. There is not a clear motivation why your study selected the four fields;
3. The Literature review doesn't clarify if there are or not previous studies regarding reviews on each field, separated and/or together. In addition, it is unclear why your study is better than these studies, if any;
4. The article does not present the research methodology applied for a good understanding of the approach. From the title, we understand that it is a review, but from the content we understand that it is a semi-structured literature review. This aspect must be clarified for a good understanding of magnitude of the research.
5. The management of figures and tables in the article is not appropriate. The reference of the figures must be exhaustive in text, the order of the tables must be 1, 2, 3, 4, and not 4, 3, 2, 1. The quality of the figures is low. For each figure and table must specify if this is original or is adapted.
6. Some problems are regarding the references in text and at the end, but this is a formal aspect.
7. There is no discussion about the limitation of this study.
8. The Conclusion is not appropriate for this study. Please, respect the structural standard for this section.
9. The references at the end must respect the standard of MDPI Journals.
Author Response
Dear Reviewer,
Thank you for reviewing our article. Your suggestions have tremendously upgraded the readability and quality of my paper. I am grateful for your recommendations and valuable comments.
- The abstract must contain very clearly the objective, the methodology applied, the results on every four fields, and originality. Thank you for your valuable comment.
Response: The abstract had been updated based on the reviewer’s comments
- There is not a clear motivation why your study selected the four fields;
Response: Thank you for your comments.
You are right. This paper selected these 4 fields as critical
- The Literature review doesn't clarify if there are or not previous studies regarding reviews on each field, separated and/or together. In addition, it is unclear why your study is better than these studies, if any;
No previous studies review the fields of the industry together. This paper will contribute to the body of knowledge about big data applications in these fields. Moreover, this paper reviewed the existing literature to compare the Big Data applications before and after COVID-19. The comparison was supported with examples from four vital fields in the industry of Healthcare, Education, Transportation, and Banking as most sectors are affected by COVID-19. The paper presented a detailed description of the role of data analytics and its alignment with specific big data applications in those fields. Such applications supported organizations and nations to navigate through the COVID-19 pandemic confidently.
Response: Thank you for your valuable comment.
- The article does not present the research methodology applied for a good understanding of the approach. From the title, we understand that it is a review, but from the content we understand that it is a semi-structured literature review. This aspect must be clarified for a good understanding of magnitude of the research.
Response: Thank you for your valuable comment to improve our paper. The manuscript had been updated in the abstract and introduction to show that this paper is review paper.
- The management of figures and tables in the article is not appropriate. The reference of the figures must be exhaustive in text, the order of the tables must be 1, 2, 3, 4, and not 4, 3, 2, 1. The quality of the figures is low. For each figure and table must specify if this is original or is adapted.
Response: The manuscript had been updated based on your comments.
- Some problems are regarding the references in text and at the end, but this is a formal aspect.
Response: We tried the best to revise the references as we used a reference tool
- There is no discussion about the limitation of this study.
Response: The limitation had been added. The limitation of this paper is related to the limited previous studies that investigated the applications and opportunities of big data during the COVID-19 Pandemic.
- The Conclusion is not appropriate for this study. Please, respect the structural standard for this section.
Response: The conclusion had been updated based on your comments.
- The references at the end must respect the standard of MDPI Journals.
Response: We tried the best to revise the references as we used a reference tool
Kindly find the response to the comments you suggested in our paper.
Thank you so much,
Reviewer 3 Report
1. The framework was explored alongside the Big data analytics compact process flow.
2. The ideas of Big Data Analytics Applications and Opportunities, as well as their relevant concerns, have been thoroughly described.
3. This study adds to the development of new methods for future algorithms by utilising Big data in the form of ultimate big data apps and their related data analytics to produce an empirical rule on Healthcare-based applications.
4. The proposed study, with its empirical rule, offers a very high precision value, where a beneficial and practicable path that firms can follow to obtain superior Analytics is presented in the most straightforward manner possible.
5. Can the provided study achievements be analysed for any other key development aspect?
6. In addition to Big Data approaches, it may benefit from a few other crucial features, such as the construction of an interpretable model with visualisation.
7. Highlight any minor obstacles that can be overcome.
8. It is preferable to ensure that the performance metrics for this set of differentiating limited datasets have been thoroughly validated before conducting temporal analysis.
At the moment, authors claim that "For instance, in the education sector, big data analytics played a key role in overcoming the negative consequences of the pandemic on the educational sector. It supported tutors and instructors to personalize the remote learning experience for educators." Line 64 and "However, this paper presents new insights into how big data analytics is integrated into the picture in four critical industries. More specifically, this paper will present how big data applications and their aligned data analytics can pave the road for industries to survive an uncontrolled and unpredictable situation. " Line 77 Below papers have some interesting implications that you could discuss in your introduction/literature review and explain it relating to your work.
N. Singh, V. K. Gunjan, G. Chaudhary, R. Kaluri, N. Victor, and K. Lakshmanna, “IoT enabled HELMET to safeguard the health of mine workers,” Comput. Commun., vol. 193, pp. 1–9, Sep. 2022.
Mohammed Usman, Vinit Kumar Gunjan, Mohd Wajid, Mohammed Zubair, Kazy Noor-e-alam Siddiquee, "Speech as a Biomarker for COVID-19 Detection Using Machine Learning", Computational Intelligence and Neuroscience, vol. 2022, Article ID 6093613, 12 pages, 2022. https://doi.org/10.1155/2022/6093613
Belmon, A.P., Auxillia, J. (2022). IoT-Based Continuous Glucose Monitoring System for Diabetic Patients Using Sensor Technology. In: Satyanarayana, C., Gao, XZ., Ting, CY., Muppalaneni, N.B. (eds) Machine Learning and Internet of Things for Societal Issues. Advanced Technologies and Societal Change. Springer, Singapore. https://doi.org/10.1007/978-981-16-5090-1_3
Author Response
Dear Reviewer,
Thank you for reviewing our article. Your suggestions have tremendously upgraded the readability and quality of my paper. I am grateful for your recommendations and valuable comments.
- The framework was explored alongside the Big data analytics compact process flow
Response: Thank you so much.
- The ideas of Big Data Analytics Applications and Opportunities, as well as their relevant concerns, have been thoroughly described.
Response: Thank you so much.
- This study adds to the development of new methods for future algorithms by utilising Big data in the form of ultimate big data apps and their related data analytics to produce an empirical rule on Healthcare-based applications.
Response: Thank you so much.
- The proposed study, with its empirical rule, offers a very high precision value, where a beneficial and practicable path that firms can follow to obtain superior Analytics is presented in the most straightforward manner possible.
Response: Thank you so much.
- Can the provided study achievements be analysed for any other key development aspect?
Response: Thank you so much. Yes sure, future work for us as Systematic Literature Review will investigate that in another key development aspect like media and communications, etc….
- In addition to Big Data approaches, it may benefit from a few other crucial features, such as the construction of an interpretable model with visualisation.
Response: Thank you so much for your valuable comment. The characteristic of visualizations had been highlighted as a characteristic of Big Data.
“Another work done by [53], defined the 7 Vs of BD namely Volume, Velocity, Variety, Value, Veracity, Variability which implies inconstancy and heterogeneity; and visualization which implies the illustrative character of data. Volume, Velocity, Variety, Veracity, and Value are the widely accepted and common Vs by stakeholders”
- Highlight any minor obstacles that can be overcome.
Response: This review paper highlights the challenges of Big Data based on the characteristic of Big Data
“The META Group analyst Dough Laney (now Gartner) has defined data growth challenges and opportunities in to three-dimensional (velocity, volume, variety) [18], [28]”
“The future work will start by investigating the challenges faced by organizations on different levels, it will also investigate the critical success factors of Big Data and their categories toward developing a conceptual model for Big Data implementation. “
Future work will investigate the critical challenges of Big Data Analytics and applications during the COVID-19 pandemic.
- It is preferable to ensure that the performance metrics for this set of differentiating limited datasets have been thoroughly validated before conducting temporal analysis.
Response: Thank you. Future work will validate the performance metric using an interview with experts to develop a model of CSFs.
Kindly find the response to the comments you suggested in our paper.
Thank you so much,
Reviewer 4 Report
The Figure 1 resolution is too bad for such a nice picture, especially letters that look bad, improve the quality, please. The overall overview of applications spectre is nice.
However, you described these V's characterising big data without any further application in your article. The question is: in which cases the Volume is the problem, where is Velocity, where Variety?
Another important thing which I missed in this article is the integration of systems which are used in BigData solutions into the current review. You can't overview Big Data without considering systems, or at least their types (groups). In fact, Big Data definition was originally given birth by these systems, you can simply read it as "the problems which must be solved with these systems", the emerging wave of technological solutions and the need for them before they appeared needed a name, so that is how "BigData" term was born. The classical multiple V's definition is the consequence, however even with these V's presented, to answer the question of whether something is Big Data or not you need to observe the need for BigData systems - if the problem doesn't need a Big Data system it shouldn't be considered a Big Data problem.
You have that "Method/ Model" in the table. However, what is really important for the reader is the knowledge about which systems nowadays cover which models. Also, your "Method/ Model" refers to actual applications or models in problem formulations without direct connection Big Data itself, rather than models (see e.g. Belcastro, L., Marozzo, F., Talia, D., & Trunfio, P. (2017). Big data analysis on clouds. In Handbook of big data technologies (pp. 101-142). Springer, Cham.) ... classical big data (programming) models are: MapReduce, DAG, MP, BSP, Workflow, SQL-like and so on. I wonder if there are any new ones - that is what I as a reader want to see under the flag "Big Data Analytics Applications and Opportunities".
Next, there are systems, implementing those models: Apache Hadoop, Spark, Storm, Flink, Giraph, MPI and so on ... what is interesting to me is to see what are the current new opportunities and why they are good. You must somehow discuss which models and systems are used in which cases, otherwise, for me it looks too abstract to be useful and to reflect the name of the article.
Author Response
Dear Reviewer,
Thank you for reviewing our article. Your suggestions have tremendously upgraded the readability and quality of my paper. I am grateful for your recommendations and valuable comments.
The Figure 1 resolution is too bad for such a nice picture, especially letters that look bad, improve the quality, please. The overall overview of applications spectre is nice.
However, you described these V's characterising big data without any further application in your article. The question is: in which cases the Volume is the problem, where is Velocity, where Variety?
Response: Thank you so much for your valuable comment. Figure 1 had been updated based on your comment.
Another important thing which I missed in this article is the integration of systems which are used in BigData solutions into the current review. You can't overview Big Data without considering systems, or at least their types (groups). In fact, Big Data definition was originally given birth by these systems, you can simply read it as "the problems which must be solved with these systems", the emerging wave of technological solutions and the need for them before they appeared needed a name, so that is how "BigData" term was born. The classical multiple V's definition is the consequence, however even with these V's presented, to answer the question of whether something is Big Data or not you need to observe the need for BigData systems - if the problem doesn't need a Big Data system it shouldn't be considered a Big Data problem.
You have that "Method/ Model" in the table. However, what is really important for the reader is the knowledge about which systems nowadays cover which models. Also, your "Method/ Model" refers to actual applications or models in problem formulations without direct connection Big Data itself, rather than models (see e.g. Belcastro, L., Marozzo, F., Talia, D., & Trunfio, P. (2017). Big data analysis on clouds. In Handbook of big data technologies (pp. 101-142). Springer, Cham.) ... classical big data (programming) models are: MapReduce, DAG, MP, BSP, Workflow, SQL-like and so on. I wonder if there are any new ones - that is what I as a reader want to see under the flag "Big Data Analytics Applications and Opportunities".
Next, there are systems, implementing those models: Apache Hadoop, Spark, Storm, Flink, Giraph, MPI and so on ... what is interesting to me is to see what are the current new opportunities and why they are good. You must somehow discuss which models and systems are used in which cases, otherwise, for me it looks too abstract to be useful and to reflect the name of the article.
Response: The Big Data Challenges related to their characteristics are situational.
The manuscript had been updated to show that. “Big data is more of a concept than an exact term. Some classify big data as a volume problem only for petabyte-scale (>1 million GB) data collection. Some people associate big data with different data types, even if the volume is measured in terabytes. These interpretations made the big data problem situational [53].“
You are right. A systematic literature review and A comparative analysis for the existing Big Data Systems and models will be investigated as future work on our new ongoing paper under writing.
Kindly find the response to the comments you suggested in our paper.
Thank you so much,
Round 2
Reviewer 3 Report
> I had suggested to broaden the Literature review as it's too small
> and narrow which is not addressed. I suggest to include the works
> done in the Literature review by relating it to their work
>
> Ulaga Priya, K., Pushpa, S. (2022). Resampling Imbalanced Data and
> Impact of Attribute Selection Methods in High Dimensional Data. In:
> Satyanarayana, C., Gao, XZ., Ting, CY., Muppalaneni, N.B. (eds)
> Proceedings of the International Conference on Computer Vision, High
> Performance Computing, Smart Devices and Networks. Advanced
> Technologies and Societal Change. Springer, Singapore.
> https://doi.org/10.1007/978-981-19-4044-6_2
>
> Akula, C.S., Prathima, C., Srinivasulu, A. (2022). Reliable Smart
> Grid Framework Designs Through Data Processing and Analysis Process.
> In: Satyanarayana, C., Gao, XZ., Ting, CY., Muppalaneni, N.B. (eds)
> Proceedings of the International Conference on Computer Vision, High
> Performance Computing, Smart Devices and Networks. Advanced
> Technologies and Societal Change. Springer, Singapore.
> https://doi.org/10.1007/978-981-19-4044-6_20
>
> Merugu, S., Kumar, A., Ghinea, G. (2023). Hardware, Component,
> Description. In: Track and Trace Management System for Dementia and
> Intellectual Disabilities. Advanced Technologies and Societal
> Change. Springer, Singapore.
> https://doi.org/10.1007/978-981-19-1264-1_5
>
> Das, T., Mukherjee, S. (2022). Data Privacy in IoT Network Using
> Blockchain Technology. In: Mukherjee, S., Muppalaneni, N.B.,
> Bhattacharya, S., Pradhan, A.K. (eds) Intelligent Systems for Social
> Good. Advanced Technologies and Societal Change. Springer, Singapore.
> https://doi.org/10.1007/978-981-19-0770-8_10
>
Author Response
Dear reviewer. Thank you for your comment.
The suggested papers have been added.
Reviewer 4 Report
In comments about Table 1 you use the term "big data analytics model", and I disagree with that. Machine learning doesn't automatically mean big data. Maybe you should change it to "data processing models used to analyse big data", because again (you ignored my comment about big data methods/models before), these models are just data analysis(processing) models, but not big data processing (they might be used for that, though).
I agree with a lot when you say "Big data is more of a concept than an exact term", that is why I mentioned big data systems in my previous response. You can't convince me that something is big data unless you needed a big data solution using systems which are considered by experts as big data systems. If you need only a single computer with machine learning Python libraries for a solution - it is not Big Data for sure. Thus, predictive analytics are not "big data analytics model" but can be applied to process big data. You must clarify it and change your terms usage.
As for your response "You are right. A systematic literature review and A comparative analysis for the existing Big Data Systems and models will be investigated as future work on our new ongoing paper under writing."
You must explicitly say that to the reader in the first place, not to me. Moreover, it is necessary to consider Big Data in the big data systems context to make a full picture of opportunities and current trends. It is a current flaw ("todo") of this research so show it to the reader, it must be transparently visible in your article.
Author Response
Dear Reviewers,
Thank you for reviewing our article. Your suggestions have tremendously upgraded the readability and quality of my paper. I am grateful for your recommendations and valuable comments.
In comments about Table 1 you use the term "big data analytics model", and I disagree with that. Machine learning doesn't automatically mean big data. Maybe you should change it to "data processing models used to analyse big data", because again (you ignored my comment about big data methods/models before), these models are just data analysis(processing) models, but not big data processing (they might be used for that, though).
- You are right. We benefited a lot from your experience and comment. I updated the methods/models to “data processing models used to analyze big data” as you requested and thanks for convincing us about that truth.
I agree with a lot when you say "Big data is more of a concept than an exact term", that is why I mentioned big data systems in my previous response. You can't convince me that something is big data unless you needed a big data solution using systems which are considered by experts as big data systems. If you need only a single computer with machine learning Python libraries for a solution - it is not Big Data for sure. Thus, predictive analytics are not "big data analytics model" but can be applied to process big data. You must clarify it and change your terms usage. big data” as you requested and thanks for convincing us about that truth.
- You are right, sorry for missing those predictive analytics is not a "big data analytics model”, it is a type of data analytics that can be applied to process big data. We modified the concept in the table. I hope we did the update as requested by you.
As for your response "You are right. A systematic literature review and A comparative analysis for the existing Big Data Systems and models will be investigated as future work on our new ongoing paper under writing." You must explicitly say that to the reader in the first place, not to me.
- You are right. I updated the abstract to show to the reader our future work.
“Future work will conduct a systematic literature review and a comparative analysis of the existing Big Data Systems and models. Moreover, future work will investigate the critical challenges of Big Data Analytics and applications during the COVID-19 pandemic. “
Moreover, it is necessary to consider Big Data in the big data systems context to make a full picture of opportunities and current trends. It is a current flaw ("todo") of this research so show it to the reader, it must be transparently visible in your article.
- Many thanks. The manuscript had been updated.
“There is undoubtedly a need for the transfer/exchange the big data systems since decision-makers must be able to react swiftly to changes or trends in markets, investments, interest rates, and other crucial happenings [3]. Decision-makers should be thoroughly aware of the type of inputs they have and the best structure for exchange or analysis if they are thinking about making significant investments in KM systems or big data/business analytics systems[3]. “
“Organizations from a variety of industries have started using MapReduce-based solutions for processing enormous amounts of data [71]. To meet their needs for handling large-scale data processing, many businesses rely on MapReduce. As businesses from a variety of sectors embrace MapReduce together with parallel databases. new MapReduce workloads have appeared that contain a large number of brief, interactive tasks [71].”
Thank you so much,